Architecture design of a reinforcement environment for learning sign languages

Naranjo-Zeledón Luis 1 2 lcnz@alu.ua.es
http://orcid.org/0000-0002-8857-162X Chacón-Rivas Mario 1
http://orcid.org/0000-0003-1537-0218 Peral Jesús 2
Ferrández Antonio 2
1 Inclutec, Costa Rica Institute of Technology , Cartago , Costa Rica
2 Department of Languages and Computing Systems, University of Alicante , Alicante , Spain
Zhao Wenbing
Electronic publication date: 2021 Oct 12
Publication date: 2021
Volume: 7
Electronic Location ID: e740
Received 2021 Apr 28; Accepted 2021 Sep 17
Copyright: © 2021 Naranjo-Zeledón et al.
Copyright year: 2021
Copyright holder: Naranjo-Zeledón et al.
License: This is an open access article distributed under the terms of the Creative Commons Attribution License, which permits unrestricted use, distribution, reproduction and adaptation in any medium and for any purpose provided that it is properly attributed. For attribution, the original author(s), title, publication source (PeerJ Computer Science) and either DOI or URL of the article must be cited.
License URL: https://creativecommons.org/licenses/by/4.0/

Keywords: Sign Language, Phonological proximity, Learning reinforcement, Similarity measures

Funding: Spanish Ministry of Science, Innovation and Universities RTI2018-094283-B-C32 Spanish Ministry of Science and Innovation PID2020-112540RB-C43 Project INTEGER RTI2018-094649-B-I00 Conselleria de Educación, Investigación, Cultura y Deporte of the Community of Valencia, Spain PROMETEO/2018/089 School of Computing and the Computer Research Center at Costa Rica Institute of Technology and CONICIT (Consejo Nacional para Investigaciones Científicas y Tecnológicas), Costa Rica 290-2006 This work was supported by the Spanish Ministry of Science, Innovation and Universities through the Project ECLIPSE-UA under Grant RTI2018-094283-B-C32, the Spanish Ministry of Science and Innovation through the Project AETHER-UA under Grant PID2020-112540RB-C43, the Project INTEGER under Grant RTI2018-094649-B-I00, and by the Conselleria de Educación, Investigación, Cultura y Deporte of the Community of Valencia, Spain, within the Project PROMETEO/2018/089, the School of Computing and the Computer Research Center at Costa Rica Institute of Technology and CONICIT (Consejo Nacional para Investigaciones Científicas y Tecnológicas), Costa Rica, under grant 290-2006. The funders had no role in study design, data collection and analysis, decision to publish, or preparation of the manuscript.

==============================
Different fields such as linguistics, teaching, and computing have demonstrated special interest in the study of sign languages (SL). However, the processes of teaching and learning these languages turn complex since it is unusual to find people teaching these languages that are fluent in both SL and the native language of the students. The teachings from deaf individuals become unique. Nonetheless, it is important for the student to lean on supportive mechanisms while being in the process of learning an SL. Bidirectional communication between deaf and hearing people through SL is a hot topic to achieve a higher level of inclusion. However, all the processes that convey teaching and learning SL turn difficult and complex since it is unusual to find SL teachers that are fluent also in the native language of the students, making it harder to provide computer teaching tools for different SL. Moreover, the main aspects that a second language learner of an SL finds difficult are phonology, non-manual components, and the use of space (the latter two are specific to SL, not to spoken languages). This proposal appears to be the first of the kind to favor the Costa Rican Sign Language (LESCO, for its Spanish acronym), as well as any other SL. Our research focus stands on reinforcing the learning process of final-user hearing people through a modular architectural design of a learning environment, relying on the concept of phonological proximity within a graphical tool with a high degree of usability. The aim of incorporating phonological proximity is to assist individuals in learning signs with similar handshapes. This architecture separates the logic and processing aspects from those associated with the access and generation of data, which makes it portable to other SL in the future. The methodology used consisted of defining 26 phonological parameters (13 for each hand), thus characterizing each sign appropriately. Then, a similarity formula was applied to compare each pair of signs. With these pre-calculations, the tool displays each sign and its top ten most similar signs. A SUS usability test and an open qualitative question were applied, as well as a numerical evaluation to a group of learners, to validate the proposal. In order to reach our research aims, we have analyzed previous work on proposals for teaching tools meant for the student to practice SL, as well as previous work on the importance of phonological proximity in this teaching process. This previous work justifies the necessity of our proposal, whose benefits have been proved through the experimentation conducted by different users on the usability and usefulness of the tool. To meet these needs, homonymous words (signs with the same starting handshape) and paronyms (signs with highly similar handshape), have been included to explore their impact on learning. It allows the possibility to apply the same perspective of our existing line of research to other SL in the future.

Introduction

It is important to lean on learning reinforcement tools to enhance the learning process of a sign language (SL), the knowledge obtained from class work seems to be insufficient to meet this purpose. The use of technology becomes a great option to achieve the objective of learning a sign language because the student faces a new language based on his visual abilities rather than the spoken word. The use of technology can offer easy-to-use visual interfaces permitting the interested party to compare and associate the meaning of signs more accurately. The interested party has the possibility to access these applications if necessary.

A previous research (Naranjo-Zeledón et al., 2020) showed that phonological proximity upgrades different areas within the study of a Sign Language. In relation to this particular research, the fundamental characteristics of the Costa Rican Sign Language (LESCO, for its acronym in Spanish) were disclosed regarding phonological proximity for clustering and learning reinforcement purposes.

The main contributions of our proposal are: The design of a modular and portable architecture of a learning reinforcement environment for sign languages (it can be applied to different sign languages).

The inclusion of a graphical software tool, including a signing avatar and a phonological proximity component meant to enrich the learning process.

The identification of homonym and paronym signs that illustrate the different degrees of proximity between the signs.

The evaluation of the phonological proximity module at the implementation level, as well as the usability and usefulness of this implementation in a software tool.

The architecture design includes the aforementioned software tool, containing a phonological component so to make the learning process more complete. This tool was developed by the Costa Rica Institute of Technology. At first, the tool showed concepts classified into different categories (colors, alphabet, numbers, etc.). These classifications are grouped into three levels: basic, intermediate, and advanced. If a sign is chosen by the user, an avatar will reproduce it, interfacing at the same time with the PIELS (International Platform for Sign Language Edition, for its acronym in Spanish). So far, it displays content from LESCO, but the design of the tool allows for adaptation to any SL (Serrato-Romero & Chacón-Rivas, 2016).

Our phonological proximity study analyzes homonyms (signs with the same starting handshape), paronyms (similar signs with different meanings), and polysemy which we have determined is very rare in LESCO (same sign with different meanings). Some examples of these phenomena are easy to find in spoken languages such as the paronyms ‘affect’ and ‘effect’, or the homonyms of ‘book’ (‘something to read’ or ‘make a reservation’). Sign languages require to employ visual variables, like handshapes, their location at the pointing spot and facial gestures. The software tool was broadened to examine these correlations and their impact on reinforcing learning.

Conventional techniques in use, as will be seen in the Background section, have drawbacks because they are specifically designed for some specific sign language. This situation is very possibly since they do not use, or at least do not make explicit, a formalization of the grammar of the specific sign language. This is in direct contrast to our approach, which takes as a starting point the formalization of phonological parameters mapped to integer numbers and their subsequent use by applying similarity measures.

We emphasize the difference between other methods to better understand the position of this work. In this way, the research community can formalize their own parameters and adopt our approach without the need for any change at the architectural level. In the field at hand, which is education, the proposals revolve around the use of various techniques or approaches, each with its rationale and justification, but at the same time revealing clear disadvantages, as explained below: Self-assessment open-source software, with web-based tests for adult learners. These Yes/No tests require important improvements to offer a more complete service, such as providing the option to see translations directly after indicating whether the user knows or does not know a sign.

Using hardware devices (wearables, usually Kinect) with recognition of hand movements and guidance to learners. As a general rule, the use of wearables is preferable to be left as the last option, because it is unnatural, it is expensive and the equipment can be lost or damaged, hindering the process.

Educational games (who also use wearables). The same downsides identified in the previous point are faced here.

Incorporation of computer vision to indicate unnatural movements in the novice. While it is true that computer vision is a very fertile field of research today, but it also requires the use of additional equipment.

Teaching fingerspelling in the form of quizzes. In general, fingerspelling presents a communication technique that is too limited and should only be used when the signaling person does not have any other resources to communicate their message at all.

Lexicon teaching proposals. can be incomplete when the desired outcome is to produce real communications, with syntactic connections that make sense to the other party.

Our main objective is to demonstrate that SL learning can be reinforced by technological means incorporating the concept of phonological proximity. In turn, we will explain how the phonological components of the SL in question should be parameterized, to incorporate them into an architecture that provides a suitable interface with a signing avatar.

In conclusion, in this paper we provide elements that clarify how to deal with issues of central importance in this type of technologies, hence making clear the advantages of the proposed system, specifically: Modular architecture, to simplify the maintenance and the incorporation of new functionalities.

Applicable to several sign languages, to take advantage of the conceptual power of our contribution in other languages and other tproypes of projects.

Enrichment of learning environments, to take advantage of opportunities to accelerate and improve the experience.

Differentiation between homonyms and paronyms, to contrast the different degrees of proximity between the signs and, therefore, the need to emphasize the practice where it is most necessary.

Applicable to other environments, offering a portable concept where researchers can incorporate it and take advantage of it.

Usability and usefulness validated through a standard test, to have a high degree of certainty that the concept of proximity is properly supported by a real tool, which in turn is easy to use.

The next section provides a background of previous work on the subject matter. Then, we present the proposed architecture of the learning reinforcement environment (‘Architecture for SL learning reinforcement environment’ section). After that, we illustrate details of its deployment in a software tool showing the experiments carried out with the phonological proximity module and the users’ validation (‘The SL Learning Reinforcement Tool module’ section). Finally, we draw our conclusions, depicting our contribution to the sign language learning process and the future work (‘Conclusions’ section).

Background

This section presents the antecedents of the object of study. They have been classified into three groups of great significance, each one explained in the following subsections. First, the importance of phonological proximity is addressed, which is a crucial matter in this study. Then, the similarity measures for phonological proximity are examined, which are the computational mechanism to determine similarity between objects to be compared. Finally, we go over the proposals for teaching tools that have been made for the student to practice sign languages. We believe that based on the exhaustive search performed in scholarly repositories, this literature reflects an up-to-date state of the research on these topics.

The importance of phonological proximity

According to Baker et al. (2016) the main aspects that a second language learner of a sign language finds difficult are phonology, non-manual components, and the grammatical use of space. These features are specific to sign languages and do not take place in spoken languages. Moreover, the phonological inventory of sign languages is completely different from that of spoken languages. Another salient observation of the authors has to do with how among the different phonological parameters, the handshape has the largest number of distinctive possibilities. In the sign languages that have been studied, the number of different handshapes appears to be larger than the number of locations and movements. Iconicity is also to be considered since signs commonly portray iconic features, which means that the handshape resembles parts of the meaning, a phenomenon that is extremely rare to find in spoken languages.

Williams, Stone & Newman (2017) have studied the importance of phonological similarity to facilitate lexical access, that is, the process by which individuals produce a specific word from their mental lexicon or recognize it when it is used by others (American Psychological Association, 2022). This study, rooted in this psycholinguistic aspect, has determined that lexical access in sign language is facilitated by the phonological similarity of the lexical representations in memory.

Keane et al. (2017) have demonstrated that for fingerspelled words in American Sign Language (ASL), the positional similarity score is the description of handshape similarity that best matches the signer perception when asked to rate the phonological proximity. The positional similarity approach is superior when compared to the contour difference approach, so in order to define similarity when fingerspelling, it is more important to look at the positional configuration of the handshapes than concentrating on the transitions.

An eye-tracking study on German Sign Language (Wienholz et al., 2021) recorded eye movements of the participants while watching videos of sentences containing related or unrelated sign pairs, and pictures of a target and some unrelated distractor. The authors concluded that there is a phonological priming effect for sign pairs that share both the handshape and movement while they differ in their location. The results suggest a difference in the contribution of parameters to sign recognition and that sub-lexical features can influence sign language processing.

An experiment conducted by Hildebrandt & Corina (2002) revealed that all subjects, regardless of their previous exposure to ASL, categorize signs that share location and movement (and differ in handshape) as highly similar. However, an ulterior examination of additional parameter contrasts revealed that different degrees of previous linguistic knowledge of the signers influenced the way they perceived similarity. So, for instance, the combination of location and handshape is recognized as carrying a higher level of similarity by native signers than by late deaf learners or by hearing signers.

Similarity measures for phonological proximity

With regard to measuring the phonological proximity in a sign language, different similarity measures can be used. They can be categorized into five categories: Edit-based, Token-based, Hybrid, Structural (Domain-dependent), and Phonetic (Naumann & Herschel, 2010; Bisandu, Prasad & Liman, 2019).

Similarity measures can be used, as long as the characteristics of the data used as phonological parameters have been properly characterized. For instance, if the parameters are strings of characters, then the edit-based similarity measurements can satisfy the objective. If the parameters are token sets (this is our case), then token-based measures can be used successfully. Hybrid approaches strive for a balance regarding the response speed of other known measures and the robustness of comparison between all the tokens, so as to find the best matches, both to deal with named entities and to solve problems of misspelling in big data contexts (this fact does not point them as good candidates for sign languages). Phonetic measures, due to their very nature, have been extensively used for spoken languages, so they are not a good choice for sign languages. Finally, the domain-dependent measures use particularities of the data, which do not fit well in corpora of sign languages.

The current similarity measures have been widely welcomed by the research community and many of them are long-standing, hence the literature on the matter can be classified as classical. Coletti & Bouchon-Meunier (2019) note that a complete review or even a simple listing of all the uses of similarity is impossible. They are used in various tasks ranging from management of data or information, such as content-based information retrieval, text summarization, recommendation systems, to user profile exploitation, and decision-making, to cite only a few. Among the many similarity measures proposed, a broad classification may be of use: the classical crisp context (Choi, Cha & Tappert, 2010; Lesot, Rifqi & Benhadda, 2009) and the fuzzy context (Bhutani & Rosenfeld, 2003; Bouchon-Meunier, Rifqi & Bothorel, 1996; Li, Qin & He, 2014; Couso, Garrido & Sánchez, 2013). Due to the nature of our research, we concentrate on the classical crisp context, since the fuzzy scenario does not apply to our object of study. The main characteristics of these predominantly classic measures are: Edit-based focusing on the calculation of the changes necessary to produce one string from another, weighing the number of necessary changes (insertions, deletions or modifications) to produce the new string. Hamming distance (Hamming, 1950) and Levenshtein distance (Levenshtein, 1966) are the best known.

Token-based approaches measuring the number of matches between two sets of parameters, (n-gram tokens), where tokens are words or numbers. In this category we can mention Jaccard distance (Jaccard, 1912) and Cosine distance (Singhal, 2001).

Hybrid strategies comparing strings, using an internal similarity function (Jaro or Levenshtein, for instance). Monge–Elkan (Monge & Elkan, 1997) and Soft TF-IDF (Cohen, Ravikumar & Fienberg, 2003) are examples of these techniques.

Structural proposals focusing on data particularities (Domain-dependent). Dates (Naumann & Herschel, 2010) is the best-known example.

Phonetic measures matching similar sounds in spoken languages (for example, they give maximum qualification to pairs of words such as ‘feelings’ and ‘fillings’), applying pre-established rules of similar sounds. Soundex (Russell, 1918) and Kölner–Phonetik (Postel, 1969) follow this strategy.

Proposals for teaching tools

Currently, studies for LESCO seem to be insufficient. That is why this section goes into software-based proposals, despite the fact they do not consider this foremost linguistic concept. Also, some researchers focus on the study of sign languages they can relate to more easily, be it their own country or workplace sign language in use; this situation is very common in the research community. Because of this situation, it seems relevant to mention the authors findings and proposals in this section. The education for deaf people or interpreters of sign language is out of the scope of this research. Our main focus stands for reinforcing the learning process for hearing people as final users.

In Haug & Ebling (2019) a report about the use of open-source software for sign language learning and self-assessment has been made. Another finding has been the web-based test for Swiss German Sign Language (DSGS, for its acronym in Swiss German), designed for adult learners. They gave important feedback on the appropriateness of the DSGS vocabulary self-assessment instrument. This feedback arose inputs for the system improvement. The innovation of this study relies on the fact about using existing open-source software as a starting point to develop and evaluate a DSGS test for self-assessment purposes.

The target of the study in Álvarez-Robles, Álvarez & Carreño-León (2020) is to bring forward an interactive software system (ISS) making use of a hardware device (Leap Motion) so that average users had fluid communication with deaf people. The objective is to allow a natural recognition of hand movements by helping the average users learning Mexican Sign Language (MSL) at the same time. Also, through gamification techniques, it would permit the user to learn and communicate with deaf people.

Related to British Sign Language (BSL), there is a lack of games available in the marketplace for teaching purposes indicating that minimal efforts have been made to meet this objective (Kale, 2014). The intention of the study is to develop a prototype using the Microsoft Kinect, to help teachers educate young students. This prototype would teach basic BSL, by using JavaScript and HTML5 in a web browser. Positive feedback coming from interviews and playtests among ten sign language experts unfamiliar with games technology to teach sign language, was also collected. They indicated that this prototype could be used as complement for those conventional teaching methods.

Another research (Huenerfauth et al., 2016) revealed there is a lack of interactive tools for those students learning ASL. These tools might provide them feedback on their signing accuracy, whenever their ASL teacher is not available for them. A software system project was also performed by utilizing a Kinect camera. By incorporating computer vision, this software can identify aspects of signing by showing non-natural movements so to provide feedback to the students in their won practice. This tool is not supposed to replace feedback from ASL teachers. However, the tool can detect errors. Students state it is better for them to have tools able to provide feedback like videos helping with error minimization, mainly time-aligned with their signing.

Learning sign language is a task, commonly performed in peer groups, with few study materials (Joy, Balakrishnan & Sreeraj, 2019). According to their opinion, fingerspelled sign learning turns into the initial stage of sign learning used when there is no corresponding sign, or the signer is not aware of it. Since most of the existing tools are costly because of the external sensors they use, they suggested SignQuiz, a low-cost web-based fingerspelling learning application for Indian Sign Language (ISL), with automatic sign language recognition. This application has been the first endeavor in ISL for learning signs using a deep neural network. The results reveal that SignQuiz is a better option than printed medium.

There is another available proposal for Chinese Sign Language, by Chai et al. (2017). They indicate that using a computer-aided tool known as Sign Instructor can offer an effective and efficient learning means. Even they go far beyond stating that the intervention of human teachers is no longer needed, and that the sign language learning is highly effective showing an outstanding score, even higher than the one obtained with face-to-face learning. The system has three modules: (1) a multimodal player for standard materials, including videos, postures, figures, and text; (2) online sign recognition by means of Kinect; (3) an automatic evaluation module.

There is a proposal for a Ghanaian Interactive Sign Language (GISL) Tutor (Osei, 2012). This interactive tutor becomes the first computer-based for this sign language. It was specifically designed to teach vocabulary of Ghanaian-specific signs. Those Ghanaians who were involved with this tutor and tested it, said they would even like to have more available signs. The GISL’s Tutor main purpose is letting Ghanaian-specific signs be accessible to anyone interested in using this tool, by displaying pre-recorded lessons with the help of a computerized avatar.

We can conclude this section affirming that, to the best of our knowledge, there are no documented proposals regarding software-assisted learning of sign languages that exploit the concept of phonological proximity. After having studied the background of this topic, it is clear that this research and its corresponding proposal is pertinent, insofar it explains in detail the mechanisms that can be used to incorporate a component of phonological proximity to reinforce the learning of any sign language.

Architecture for SL Learning Reinforcement Environment

The architecture of the tool has been conceived to show a high level of modularity, as well as to separate the logic and processing aspects from those associated with the access and generation of data. Obviously, by relying on several existing components, the design must show all the interdependencies that this implies.

Figure 1 shows the general architecture of the SL learning reinforcement environment, which consists of four layers, ranging from those interacting with the users to those providing signs and processing modules reflecting logically derived similitude relations. These layers are: A graphical user interface, both for web browsers and mobile devices.

The SL Learning Reinforcement Tool.

An interface with a Phonological Proximity Sub module and a Signs and Discourses repository.

A semantic disambiguation module.

Figure 1 SL learning reinforcement environment architecture.

The user interface provides access through any device that allows running a conventional web browser or running an application on a commonly used mobile device. The users can play different roles, which is relevant in the next layer, to determine the views and actions they can carry out with the tool.

The SL Learning Reinforcement Tool can be used in web or mobile environments. It consists of a sub layer called the learning module, which in turn classifies users into three roles: user (a learner), administrator, and instructional designer. These roles exist to strike a balance between separation of duties and flexibility. The user role of the tool provides access to the practice, assessment and statistics modules. The administrator role also has access to these modules, while the instructional designer role can use the practice, assessment and lesson modules. This lesson module is where the instructional design is carried out, that is, the design of each lesson, practice and evaluation mechanisms.

The other sub layer is the Phonological Proximity Component. This sub layer is in charge of interfacing with the following architectural layers, sending requests for signs in phonological close proximity every time that any of the users from any of the mentioned roles require it, as well as reproducing signs through the avatar.

Then we find two parallel layers, actually based on the already operational PIELS platform: first the Phonological Proximity Sub module and aligned to this same level the Sign and Discourses Repository. These layers interact with each other and also with the layer of the SL Learning Reinforcement Tool, previously explained.

The Phonological Proximity Submodule is responsible for receiving requests of similar signs, using a unique sign identifier. The Top-ten Petitioner Component processes these requests and returns the ten signs with the greatest phonological proximity with respect to the one received as a parameter. To do this, a repository called PIELS Similitude Matrix is queried, in which all the signs that make up the LESCO lexicon have been pre-processed. To keep this repository updated, there is a New Signs Similitude Evaluation Component, which receives each new sign included in the PIELS Sign Database from the parallel layer and applies a similarity measure between all the signs (the measure that has worked best is the cosine formula).

The Signs and Discourses Repository contains the database with the LESCO lexicon up-to-date and a collection of discourses built through the use of the PIELS platform. It also contains the signing avatar, which is in charge of visually reproducing the signs that it receives by parameter, as well as complete discourses. Naturally, to build these discourses the previously existing signs are used or new signs may be created in its built-in editor as needed. This layer provides a mechanism to embed the visual display of the avatar in the upper layer of the SL Learning Reinforcement Tool, in the modules that so demand it: lessons, practice and instructional design.

Finally, there is the layer called Semantic Disambiguation, which is intended to be used in future work. The Disambiguation Container works through a big data Web corpus and a cognitive computing module (Naranjo-Zeledón et al., 2019). It will be used to provide additional functionality to the tool, consisting of determining semantic proximity, that is, signs whose meanings are similar, regardless of whether or not they are similar in shape.

The SL learning Reinforcement Tool Module

In this paper, we focus on describing the architecture and carefully detailing the SL Learning Reinforcement Tool module, to emphasize the feasibility of our proposal from a practical point of view and to ensure that this has been validated by a group of users with a suitable profile for this task. The use of the tool seeks to demonstrate in a tangible way that it is feasible to incorporate the concept of phonological proximity in a learning reinforcement tool serving as the basis for validating this concept with sign language learners. The tool description and the performed experiments are explained in the two following subsections.

Tool description

The interface classifies signs into five categories (alphabet, numbers, greetings, Costa Rican geography and colors), displaying a new screen when choosing one of them with a list of available signs. Sequentially, the set of signs pertain to learning levels, ranging from the simplest to the most complex ones. Figure 2 exhibits the interface once logged into the system.

Figure 2 The software basic level interface with signs classified into categories.

The adaptation of the software tool is achieved by adding a functionality when choosing a signal. Whenever the user clicks on it, a list of its paronyms is shown, so the avatar reproduces the sign, then the user can request to reproduce the similar signs. In this way, the small differences can easily be determined, fact that alerts the user about how careful they should be when having a conversation with a deaf person.

Figure 3 exhibits the graphical display once the alphabet lesson module is chosen and clicked on ‘Letter-D’. The system shows the top-ten similar signs, the first sign displays a homonym ‘Where’ (from the starting handshape), and the rest of them are paronyms: ‘Letter-K’, ‘Desamparados’ (the name of a crowded city), ‘Sunday’, ‘Dangerous’, ‘Dog’, ‘Mouse’, ‘Nineteen’, ‘And’, ‘Ministry/Minister’. If the user clicks on the suggested signs, then the avatar reproduces them. Figure 4 shows the avatar pointing at ‘Letter-D’, while Fig. 5 shows paronym ‘Sunday’.

Figure 3 Top-ten homonyms/paronyms for ‘Letter-D’.

Figure 4 Avatar signing homonym ‘Letter-D’.

Figure 5 Avatar signing paronym ‘Sunday’.

When practicing, the system displays the avatar making a sign. This task can be repeated at will. Then, the student chooses the right word corresponding to the four listed options. Figure 6 illustrates this stage.

Figure 6 Avatar signing ‘Letter-U’ for practice sessions.

The system guides the student’s learning process by following a sequence of steps. This feature makes possible for the student to take lessons, save his progress status and, have access to new levels within the application to learn new concepts. The system also allows the theme to be changed, so to increase contrast levels facilitating accessibility for low-vision individuals. Besides, the student can access statistics to measure his daily progress.

Experimentation

The objectives of our experimentation consisted of evaluating the similarity measure used in the proposal and the validation of the tool. Therefore, the methodology was oriented to achieve these objectives. First, each sign was defined using 26 phonological parameters (13 for each hand), and then a similarity formula was applied to compare each pair of signs. These previous calculations were used to display each sign and its top-ten similar signs. Finally, the SUS usability standard test and an open qualitative question were applied, as well as a numerical evaluation to a group of students, to validate the proposal.

As will be seen later, in the user validation subsection, user research is a method that naturally fits the type of research we are presenting. User research is meant to represent a strong foundation for design decisions and general strategy. It helps in creating high-quality products for end-users, with the necessary data to back the strategy and design decisions. It also helps to identify early adopters of the product, hence discovering people who can give contextual feedback from the early stages of development.

In the two following subsections, we explain the experiments carried out to evaluate the similarity measure we have used in our phonological proximity module, and the validation with different users regarding the usability and usefulness of the tool.

Phonological proximity module

A similar measure, applied to the phonological parameters of the signs, was implemented. It produces a list ordered from highest to lowest for each one of them. To interact with the students, whenever they select a sign, the system just displays the ten most similar results to avoid overwhelming with huge outcomes Table 1 shows those similar scores gotten from sign ‘Letter-D’. As per the standard cosine formula, a higher number stands for higher similarity. As a matter of fact, if a 100 score is obtained, it means that the signs are homonyms. We also present the 26 phonological parameters for sign ‘LETTER-D’ and its top-ten similar signs, those matching parameters pop out in bold. As depicted in previous research (Naranjo-Zeledón et al., 2020), the results achieved by the cosine formula are based on mapping the phonological parameters of each sign to an array of numbers. These numbers are predefined and show different phonological characteristics of the signs like the hand orientation, the handshape, and the hand spatial location. Therefore, this standard formula is used to measure proximity among the orderings over the n-dimensional space. The formula is shown in Eq. (1).

(1) cos(x,y)=x⋅y||x|||y||

Table 1 Phonological parameters for sign ‘LETRA-D’ and top-ten similar signs.

LETTER-D	1	4	4	4	1	3	1	1	1	1	1	1	4	3	2	3	3	2	2	15	1	3	4	22	1	2		
																											Similarity	
WHERE	1	4	4	4	1	3	1	1	1	1	1	1	4	3	2	3	3	2	2	15	1	3	4	22	1	2	100	
LETTER-K	1	3	2	2	1	3	1	1	1	1	1	1	4	3	2	3	3	2	2	15	1	3	4	22	1	2	99.48	
DESAMPARADOS	1	4	4	4	1	2	1	1	1	1	1	1	1	3	2	3	3	2	2	15	1	3	4	22	1	2	99.42	
SUNDAY	1	4	4	4	1	2	1	1	1	1	1	1	4	3	2	3	3	2	1	12	2	3	4	22	1	2	99.4	
DANGEROUS	1	3	2	2	1	3	1	1	1	1	1	1	3	3	2	3	3	2	2	15	1	2	4	22	1	2	99.38	
DOG	3	3	3	3	1	4	1	1	1	1	1	1	5	3	3	3	3	2	2	15	1	2	4	22	1	2	99.36	
MOUSE	3	3	3	3	1	1	1	1	1	1	1	1	5	3	2	3	3	2	2	16	1	2	4	22	1	2	99.2	
NINETEEN	2	2	2	2	1	2	1	1	1	1	1	1	4	3	2	3	3	2	2	15	1	3	4	22	1	2	99.2	
AND	1	1	1	1	1	4	1	1	1	1	1	1	5	3	2	3	3	2	5	15	1	2	4	22	1	2	99.2	
MINISTER	3	3	3	2	1	3	1	1	1	1	1	1	3	3	1	3	3	2	1	15	1	2	4	22	1	2	99.2	
Note:

Text in bold means that the phonological parameters for a particular sign equal those for sign “LETTER-D”.

Here x and y are two arrays, both of them containing 26 entries that map 13 phonological parameters identified for each hand. If the cosine value is one, then the two arrays are identical, while a value of 0 means that they do not have anything in common.

User validation

This section is structured to first present the importance of user research, which leads to the choice of subjects. Then, we proceed to explain what these subjects were asked to do. Then, we provide a rationale for the concept of phonological proximity as the central axis of the validation process. Finally, we explain in detail how usability and utility were validated.

User research is becoming more and more relevant in the field of education and learning, which is why this research adheres to its principles, particularly for validation purposes. As Kao et al. (2018) indicated, iterative user research for products has been conducted in over 50 educational technology companies at different stages of development. User research has been used as a collaborative and interdisciplinary process gathering together experts from academic fields, teaching and learning sciences, and human-computer interaction, along with software developers, since developing effective educational products require an understanding of many expertise fields.

The profile of the subjects to whom the survey was administered consisted of 12 regular users of technological tools being involved in a process of learning a sign language, ten of them with a basic knowledge of LESCO and the other two in a more advanced level, not experts though. Their ages range from 21 to 52 years, although the average is 31 years of age, with eight men and four women. As for men, the average age is 29 and that of women is 35. A total of ten individuals out of 12 are novices, while the other two have a little more advanced knowledge of LS. Regarding their professions and academic degrees, they are classified into a Doctor of Computer Science, a Bachelor of Administration, an Industrial Design Engineer, four Computer Science undergraduates, four Computer Engineers and an Industrial Production Engineer. All of them used the web version of the tool, in order to facilitate remote interaction between the subjects and the researchers, and to more expeditiously clarify any doubts that may have arisen. We have found that when applying our proposal to these users, who are mostly novice sign language learners, the results are very satisfactory, as will be discussed later.

The subjects had to do two tests, in preparation for which they interacted directly with the tool, indicating in which cases they detected similarity between base signs and signs proposed by the system as highly similar. After performing this exercise with three base signs and their ten similar signs, they were asked to complete the SUS test, as well as answer an open question about their perception of usefulness. To complement the above, they were asked to also give a numerical rating to the utility. Although the tests were carried out remotely, it was possible to observe how the subjects decided relatively quickly if the suggested signs seemed similar or not, without having to ask the avatar to reproduce them several times.

The results obtained in the evaluation of the phonological proximity module are deeply analyzed in our previous paper (Naranjo-Zeledón et al., 2020). In our database, we have already mapped each sign to a vector of 26 numerical parameters, each one with a precise phonological meaning. The parameters follow this order: left index, left middle, left ring finger, left pinky, left finger separation, left thumb, right index, right middle, right ring finger, right pinky, right finger separation, right thumb, left rotation, left wrist posture, left interiority, right rotation, right wrist posture, right interiority, left laterality, left height, left depth, contact with the left arm, right laterality, right height, right depth, contact with the right arm.

For example, the array [1, 3, 2, 2, 1, 3, 1, 3, 2, 2, 1, 3, 3, 3, 3, 2, 3, 3, 2, 4, 15, 3, 2, 4, 15, 3, 2] contains the 26 parameters that phonologically describe the sign for “PROTECTION”, while the subarray [1, 3, 2, 2, 1, 3, 3, 3, 2, 4, 15, 3, 2] represents only the left-hand parameters. As explained above, the cosine is a token-based approach, and it was selected because it is the most natural way to handle numerical arrays of parameters, as is the case in this research. An edit-based measure of similarity would force the arrays to be converted to strings, taking care that some parameters have one digit while others have two. On the other hand, domain-dependent measures do not apply to our data, and a hybrid approach presents unnecessary complications. Obviously, phonetic similarity measures are specifically designed for spoken languages, so they are left out in this discussion.

With regard to the user evaluation, we have conducted an extrinsic evaluation through the SUS test, consisting of a questionnaire with ten items and a five Likert scale response for each option, ranging from ‘Strongly agree’ to ‘Strongly disagree’ (Brooke, 1986). Among its benefits, we can identify that it has become an industry standard, widely referenced in articles and publications. By using SUS, one can make sure of these very desirable characteristics: It is extremely easy to administer to participants.

It can be used on small sample sizes and yet attain reliable results.

It is valid in effectively differentiating between usable and unusable systems.

In addition to the SUS standardized test, we have considered it appropriate to include an open question of a qualitative nature, where the participants had to answer in a mandatory manner. The question is ‘How do you think comparing similar signs has made learning easier or more difficult for you?’. The objective of this question is to evaluate the usefulness of our tool. Then, we proceeded with a last question, to assign a numerical rating to the previous question, on a scale from 1 to 100, worded as follows: ‘Based on your answer to the previous question, how would you numerically rate the improvement in learning using similar signs? (one is the lowest, 100 is the highest)’.

To carry out the test, each participant was summoned individually and given instructions to enter the system so to become familiar with it. Then, they were asked to choose three signs corresponding to the ‘alphabet’ group and analyze the similarity with their corresponding ten most similar signs, to determine the precision threshold of the similarity formula used, as well as the possible need to refine the initial configuration of the signs before doing validations. After corroborating the levels of similarity, they proceeded to answer the SUS questionnaire, the qualitative question, and assign the numerical rating. The SUS survey format used can be seen in Fig. 7.

Figure 7 SUS (system usability scale) standard test.

Table 2 shows the results of applying the SUS test to the participants, that is, the 12 participants in the rows and the ten standard questions in the columns. There is an additional column on the far right, which corresponds to the numerical evaluation of the usefulness of the phonological proximity as perceived by the participants. Regarding the tone of the responses to the qualitative question, Table 3 shows the opinions and their tone for each participant in the study. This tone has been established by the authors as negative, mainly negative, neutral, mainly positive, or positive. The next section provides a broader discussion of the findings presented in both tables.

Table 2 SUS test and phonological proximity scores.

Participant	q1	q2	q3	q4	q5	q6	q7	q8	q9	q10	SUS
score	Proximity
score	
p1	4	1	5	1	4	1	5	1	5	1	95.0	70	
p2	5	3	5	4	2	3	4	1	3	4	60.0	90	
p3	5	1	4	1	3	2	5	3	5	1	85.0	70	
p4	3	1	5	2	5	1	5	1	5	1	92.5	100	
p5	5	1	4	1	5	1	5	1	4	1	95.0	100	
p6	5	1	5	2	4	1	5	1	5	2	92.5	90	
p7	5	1	5	1	4	1	4	1	5	3	90.0	100	
p8	5	1	5	4	4	1	5	1	5	1	90.0	100	
p9	4	1	5	1	5	1	5	1	5	1	97.5	90	
p10	4	4	3	2	5	1	5	1	4	1	80.0	100	
p11	4	1	5	1	5	4	5	1	5	1	90.0	90	
p12	4	1	5	2	5	1	5	1	5	1	95.0	90	
										Average:	89.0	91.0	
Note:

The average SUS score is 89%, while the proximity score is 91% (indicated in bold).

Table 3 Opinion/Judgment of learning improvement by using phonological proximity.

Participant	Opinion/Judgment	Tone	
p1	‘It helps me to locate possible signs with which I could be confused or improve the context of use’.	Positive	
p2	‘To be able to distinguish signs that can be confused at the time of a conversation’.	Positive	
p3	‘It seems very valuable to me, since with the same form, several signs can be practiced, then the hand becomes more skilled’.	Positive	
p4	‘I believe that having similar signs can make it easier for me to learn new signs, as long as I am interested in reviewing and comparing them thoroughly. Once you have mastered a sign, making the move to a similar one is much easier’.	Positive	
p5	‘At the beginning there are words that are similar and when you see them after a long period of time, these differences are not very noticeable, so it is useful to remember precise words well’.	Positive	
p6	‘The implementation of this section is useful and does not hinder the use of the application if it generates confusion as to why some are similar but this does not affect the flow of use’.	Positive	
p7	‘Yes, I think it is quite useful because it helps to know vocabulary similar to a letter/word, and it also helps to see the differences between each one so as not to be mistaken’.	Positive	
p8	‘The implementation of this section is useful and does not hinder the use of the application. It creates confusion as to why some are alike but this does not affect the flow of use’.	Mainly positive	
p9	‘I think it has facilitated because one sign is memorized and the small differences are noticed with respect to another’.	Positive	
p10	‘It has facilitated [sic] because by presenting similar signs, it helps to make their difference’.	Positive	
p11	‘Seeing similar signs allows me to learn some other signs more quickly, since I can associate them’.	Positive	
p12	‘It seems to me that in learning signs, having similar signs at hand allows me to find a better way to express the message with the correct signs. In addition, identify the differences and be able to apply it when using a specific sign’.	Positive	

Discussion

The SUS usability standard test gives a score of 89, which indicates that the tool has an extremely high level of usability, since the average of a large number of studies of this nature is 68 and from 84.1 to 100 the usability is located in the ‘A+’ percentile, which is the highest (Lewis & Sauro, 2018).

Such a high ranking also has the desirable characteristic that the product is more likely to be recommended by users to their peers. This characteristic is of particular relevance when it comes to an innovative product, which presents an important differentiation compared to the existing options, in this case in the educational field.

If we examine the extreme ratings, that is, the highest and the lowest, are removed and averaged based on the remaining ten ratings, the result is still very similar, increasing slightly from 89 to 91, which is indicative that none of them have a greater weight in the overall result. On the other hand, looking for an interpretation of these extremes, it should be noted that the lowest score was 60, and it was awarded by a subject who experienced problems during the reproduction of some signs by the avatar, due to a momentary synchronization problem on the platform, which may have negatively affected her perception. The highest score, 97.5, was given by one of the three subjects with the least experience in sign languages. In both cases it seems that the extreme ratings have a fairly predictable explanation.

On the other hand, the phonological proximity score, which reflects a numerical evaluation by each participant in a scale from 1 to 100, yields a very satisfactory average of 91.0. The obtained score demonstrates that the tool is useful for our objective of reinforcing sign language learning.

Carrying out the exercise of eliminating extreme ratings again, it can be seen the majority grant a 90 or 100 and that only two subjects gave an overall rating of 70. These are the two people who are not experts but have a little more advanced knowledge in sign languages. Again, the results make sense.

The open question showed a positive tone in practically all the answers, which can be synthesized in concepts such as “usefulness”, “detection of differences” and “reduce confusion”. It is evident that these opinions are favorable and that they in fact reflect satisfaction with respect to the improvement that students perceived when using a computer tool that incorporates the concept of phonological proximity.

Obviously, the graphic display of the tool is very useful to match the visual nature of sign languages. Although it is not the central focus of this research, it is important to highlight the fact that the appropriate graphic design and an avatar that reproduces the signs as similar as possible to what the students have learned in class is decisive for the proposal as a whole to be successful.

We are of the opinion that there is room for improvement in terms of the similarity of some signs that did not seem to represent a contribution for the majority of subjects. The phonological parameterization and the formula used work well for most cases, but in some particular cases it may be that the rotation and location of the hand account for most of the similarity, leaving aside the handshape, which is precisely what the novice student looks at first. There was only one sign that gave this problem repeatedly, but it deserves attention in future work. Validation was helpful in raising this possibility.

The general appreciation that we obtain from what it is stated in this analysis of results is that both the tool used and the concept itself of phonological proximity to detect slight differences have received the endorsement of the subjects of this study. Both from a quantitative and qualitative point of view, the subjects who collaborated in the validation show a clear acceptance of phonological proximity as a valuable concept to help reinforce their learning.

Conclusions

This paper presents an architecture to strengthen the sign language learning making use of phonological proximity concept to improve results. As far as we know, this is the first time this approach has been suggested in relation to reinforce the learning of sign languages and, particularly applied to the Costa Rican Sign Language (LESCO, for its Spanish acronym). The main contributions of our proposal are: (1) a modular architecture meant to reinforce different sign languages learning; (2) the inclusion of a software tool with a phonological proximity component to assist in the learning process; (3) the identification of homonyms and paronyms to contrast the proximity levels between the signs; (4) a thorough evaluation of our phonological proximity module and the usability and usefulness of the tool.

We describe the operation of our software tool with a graphical interface that classifies concepts and reproduces those selected signs, broadening its current functionality through phonologically similar signs, in other words, with similar handshapes. To meet this purpose, we explore the incorporation of homonymous and paronyms.

By allowing the students to compare pretty similar signs, we have included the phonological proximity component into an existing tool with a suitable interface. The value of this improvement is to reduce the number of mistakes of those similar signs in a real conversation with a deaf person since this can seriously impact the communication and hinder the understanding between each other. A mapping of the signs with the other signs becomes an essential requirement to dispose of the available lexicon in advance. To ease the use, we decided that the interface should list only ten similar signs, so the user might not be confused, situation which would be counterproductive.

We evaluated both the phonological proximity module and the usability and usefulness of the tool. Thanks to the practical application of the concept of phonological proximity, learning is reinforced, as has been validated through our experimentation with sign language students in this research. We conclude that the incorporation of the phonological proximity concept to this software tool can upgrade the reinforcement of LESCO learning, offering the possibility of using the same approach of our line of research in other sign languages.

For future work, the use of standardized questionnaires for user experience, such as AttrakDiff or UEQ (User-Experience Questionnaire), along with the SUS test, are to be considered, encompassing a comprehensive evaluation of the tool. We will also study the effect of using the similarity measure on separate phonological components of signs. This would consist of validating the handshape, orientation and location of the hand as separate components, to determine if this produces improvement in the results. Additionally, we will consider the inclusion of facial gestures, head and trunk movements as possible elements that improve the accuracy of the similarity.

Our colleague at Inclutec, Víctor Romero, provided valuable access to data and feedback on the use of various tools. We also appreciate the great support received by Lery Sánchez, Daniel Martínez, Alfredo Campos, and Ana Zúñiga, at Costa Rica Institute of Technology for providing the software code upon which we built the phonological proximity component. The help of Mario Chacón-Campos to make the definitive deployment of the software on the Web was invaluable.

Additional Information and Declarations

Competing Interests

Author Contributions

Data Availability

The authors declare that they have no competing interests.

Luis Naranjo-Zeledón conceived and designed the experiments, performed the experiments, analyzed the data, performed the computation work, prepared figures and/or tables, and approved the final draft.

Mario Chacón-Rivas conceived and designed the experiments, performed the experiments, analyzed the data, authored or reviewed drafts of the paper, was in charge of managing the research project resources, and approved the final draft.

Jesús Peral analyzed the data, prepared figures and/or tables, authored or reviewed drafts of the paper, was in charge of managing the research project resources, and approved the final draft.

Antonio Ferrández analyzed the data, authored or reviewed drafts of the paper, was in charge of managing the research project resources, and approved the final draft.

The following information was supplied regarding data availability:

Experimental data along with calculations and validations are available at Zenodo: Luis Naranjo-Zeledón. (2021). Phonological proximity calculations and validation [Data set]. Zenodo. http://doi.org/10.5281/zenodo.4723991.

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
