# Peer review of "Architecture design of a reinforcement environment for learning sign languages"

_PeerJ Computer Science, doi:10.7717/peerj-cs.740_

## Round 0.1 · original submission · Major Revisions

Dear Authors,

Based on the comments received from the authors and my own observation, I recommend major revisions for the paper.

Reviewer 1 ·

Basic reporting

The abstract is NOT satisfactory because it didn't contain the following parts:
i. The importance of or motivation for the research.
ii. The issue/argument of the research.
iii. The methodology.
iv. The result/findings.
v. The implications of the result/findings.
- Relevant literature review of latest similar research studies on the topic at hand must be discussed

Experimental design

- Please cite each equation and clearly explain its terms.
- What are the evaluations used for the verification of results?

Validity of the findings

- The procedures and analysis of the data is seen to be unclear.
- The discussion is very important in research paper. Nevertheless, this section is short and should be presented completely.

Additional comments

- Please improve overall readability of the paper.

·

Basic reporting

Authors explain why they used SUS. However, they do not indicate this in the abstract. They must add it to the abstract (mentioning SUS and the open question).

Experimental design

The user validation section (Line 360) should be reorganized for better understanding. First, indicate who the users are (377-380). Second, indicate what was done by users (393-399). Finally, explain how usability was evaluated (368-376 + 385-392).

Although the process is well detailed, I suggest:
+ It would be good to know if participants tried the web or mobile version (Lines 393-399)
+ A more extensive description of the users profile should be added (i.e., age, sex, or any information to know better those who filled out the SUS). (Lines 377-380)

Validity of the findings

I suggest a more in-depth analysis of the evaluation results, i.e.,
+ is there room for improvement?
+ Was there a difference between the users? (novices vs advanced),
+ Participant 2 sus score was the lowest, why could it happen?
+ Were the participants observed? If yes, then what details did the researchers observe?

Additional comments

Authors present the architecture of a reinforcement environment for learning sign languages. They evaluate the usability of an application (based on the proposed architecture) using SUS. The app is for LESCO.

The proposal is interesting. The contributions are well detailed. Introduction and backgroud sections are fine.

In future studies, I suggest applying questionnaires for UX evaluation (i.e., attrakdiff or UEQ) combined with SUS for a comprehensive evaluation of the tool/app.

Reviewer 3 ·

Basic reporting

.

Experimental design

.

Validity of the findings

.

Additional comments

• In the Introduction section, the drawbacks of each conventional technique should be described clearly.
• Introduction needs to explain the main contributions of the work more clearly.
• The authors should emphasize the difference between other methods to clarify the position of this work further.
• The Wide ranges of applications need to be addressed in Introductions
• The objective of the research should be clearly defined in the last paragraph of the introduction section.
• Add the advantages of the proposed system in one quoted line for justifying the proposed approach in the Introduction section.

---

## Round 0.2 · Minor Revisions

The reviewer raised a few minor issues. Please address them in the revised manuscript.

Reviewer 1 ·

Basic reporting

- Make sure the Conclusion succinctly summarizes the paper. It should not repeat phrases from the Introduction!

Experimental design

- The authors should further add an explanation about the research method.

Validity of the findings

NA

Additional comments

There are only 2 studies from 2020 referred in this paper.

- Authors should add the most recent reference:

- Improved VGG Model for Road Traffic Sign Recognition

---

## Round 0.3 · accepted · Accept

The authors addressed all issues raised by the reviewer. I now recommend to accept this paper.